# *Eucheuma cottonii* Seaweed-Based Biochar for Adsorption of Methylene Blue Dye

Anwar Ameen Hezam Saeed [1,2], Noorfidza Yub Harun [1,2,*], Suriati Sufian [1], Ahmer Ali Siyal [1], Muhammad Zulfiqar [1], Muhammad Roil Bilad [1], Arvind Vagananthan [1], Amin Al-Fakih [3], Aiban Abdulhakim Saeed Ghaleb [2] and Najib Almahbashi [3]

[1] Department of Chemical Engineering, Universiti Teknologi PETRONAS, Seri Iskandar 32610, Perak, Malaysia; anwar_17006829@utp.edu.my (A.A.H.S.); suriati@utp.edu.my (S.S.); ahmer.ali_g03690@utp.edu.my (A.A.S.); muhammad_g02426@utp.edu.my (M.Z.); mroil.bilad@utp.edu.my (M.R.B.); aamafnan1993@gmail.com (A.V.)

[2] Centre of Urban Resource Sustainability, Universiti Teknologi PETRONAS, Seri Iskandar 32610, Perak, Malaysia; aiban_17004546@utp.edu.my

[3] Department of Civil and Environmental Engineering, Universiti Teknologi PETRONAS, Seri Iskandar 32610, Perak, Malaysia; amin.ali_g03663@utp.edu.my (A.A.-F.); najib_17005375@utp.edu.my (N.A.)

* Correspondence: noorfidza.yub@utp.edu.my

**Abstract:** Pollution from dye containing wastewater leads to a variety of environmental problems, which can destroy plant life and eco-systems. This study reports development of a seaweed-based biochar as an adsorbent material for efficient adsorption of methylene blue (MB) dye from synthetic wastewater. The *Eucheuma cottonii* seaweed biochar was developed through pyrolysis using a tube furnace with $N_2$ gas, and the properties were later improved by sulfuric acid treatment. The adsorption studies were conducted in a batch experimental setup under initial methylene blue concentrations of 50 to 200 mg/L, solution pH of 2 to 10, and temperature of 25 to 75 °C. The characterization results show that the developed biochar had a mesoporous pore morphology. The adsorbent possessed the surface area, pore size, and pore volume of 640 $m^2$/g, 2.32 nm, and 0.54 $cm^3$/g, respectively. An adsorption test for 200 mg/L of initial methylene blue at pH 4 showed the best performance. The adsorption data of the seaweed-based biochar followed the Langmuir isotherm adsorption model and the pseudo-second-order kinetic model, with the corresponding $R^2$ of 0.994 and 0.995. The maximum adsorption capacity of methylene blue using the developed seaweed-based biochar was 133.33 mg/g. The adsorption followed the chemisorption mechanism, which occurred via the formation of a monolayer of methylene blue dye on the seaweed-based biochar surface. The adsorption performance of the produced seaweed biochar is comparable to that of other commercial adsorbents, suggesting its potential for large-scale applications.

**Keywords:** biochar; pyrolysis; methylene blue; adsorption; seaweed

---

## 1. Introduction

A variety of harmful contaminants, such as phenols, dyes, surfactants, heavy metals, and several other personal-care-related chemicals, have been polluting water bodies [1]. Many types of dyes are harmful compounds that cause a variety of diseases and disorders, such as nausea, asthma, vomiting, blindness, and mental confusion [2,3]. Annually, 0.7 million tons of dyes are produced [4]. Dyes contained in wastewater (such as from textile processing industries) must be removed from the effluents before the treated water is discharged into water bodies. Development of eco-friendly and effective technologies for dye removal from industrial wastewater has thus become a pressing issue.

A variety of technologies can be implemented for dye removal from wastewater, such as reverse osmosis, photodegradation, adsorption, coagulation, electrochemical, biochemical degradation, and ion exchange [5]. Adsorption is considered as a very simple, economical, and effective technique for the complete removal of dyes from synthetic waste effluent [6]. The adsorbent can be made from readily available materials, such as activated carbons, bio-sorbents, rice husk ash (RHA), fly ash, zeolites, chitosan, kapok, cellulose, cotton, geopolymers, etc. [5,7–10]. Biochar has recently gained attention as an adsorbent, especially the ones derived from agricultural residues like rice husk [11,12], Kenaf [13], coconut coir [14], sawdust [15], corn straws [16], pineapple bark [17], durian husk [18], hickory wood [19], and tea waste [12]. Biochar is an effective, affordable, and environmentally friendly adsorbent material for dye removal from wastewater [20]. Adsorption onto biochar-based adsorbents derived from seaweed and algae, such as Gracilaria corticate [21], Asparagopsis armata [22], Pterocladia capillacea [23], Ascophyllum nodosum [22], and Gracilaria changii [24], has also shown positive outcomes for dye removals [25].

*Eucheuma cottonii* seaweed is abundant in countries in Asia, Africa, and Oceania [24], and has shown potential as a base material for biochar. In comparison to land crops, seaweed grows rapidly, with a high carbon dioxide fixation rate. Seaweed is readily available and abundant, and it grows well in an aquatic ecosystem [26,27]. In Malaysia, the production of seaweed was about 14,900 metric tons in 2010 and is estimated to reach about 22 metric tons in 2022 [28].

The composition of seaweed is completely different from that of land weeds. It mainly contains cellulose in the form of lignocellulosic biomass. Therefore, the seaweed is exposed to a high-energy-consumption pre-treatment step [29], which make it less attractive for conversion into bio-fuels [30,31]. The lignocellulose component is attractive as a base material for biochar adsorbent, which is addressed in this study. To the best of our knowledge, the development of seaweed-based bio-sorbents for methylene blue (MB) removal has been rarely reported, which conveys the originality of this research work.

This study investigates the development of seaweed-based biochar using the pyrolysis method for MB removal from synthetic wastewater through an adsorption technique. After fabrication and characterization, the impact of solution pH and the initial concentration of MB on the adsorption rate and capacity was assessed. Later, the isotherms and kinetics of dye adsorption were also assessed.

## 2. Experimental Details

### 2.1. Materials

*Eucheuma cottonii* seaweed was obtained from Sabah, Malaysia. The MB dye, sulphuric acid ($H_2SO_4$), hydrochloric acid (HCl), and sodium hydroxide (NaOH) were procured from Sigma-Aldrich. The chemicals were used as received without any treatment.

### 2.2. Methods

#### 2.2.1. Biochar Preparation

*Eucheuma cottonii* seaweed was pulverized using a mortar and pestle, followed by screening to the size range of 1–2 mm using a sieve shaker. Subsequently, it was washed using tap water and dried at 343 K in an electric oven for two days (48 h). The chemical activation of seaweed was carried out by putting 100 g of dry and sieved seaweed into a 10% sulfuric acid solution for 2 h. The liquid solution was then drained, and the product was continuously washed using the distilled water until all acid was removed, which was indicated by the neutral pH of the washing solution. The dried seaweed was pyrolyzed at two different temperatures of 550 and 450 °C under a $N_2$ environment purged at a flow rate of 0.0042 bar per min for 120 min [32,33].

2.2.2. Characterizations of Seaweed-Based Biochar

The standard methods of the American Society for Testing and Materials (ASTM) and a Vario Micro Element Analyzer were applied to determine both proximate and ultimate compositions of the seaweed biochar, respectively. In the proximate analysis, moisture content, volatile matter, ash content, and fixed carbon were determined using ASTM D7582-10 methods, while in the ultimate analysis, as the contents of carbon (C), nitrogen (N), hydrogen (H), sulphur (S), oxygen (O), H/C, and O/C were determined using the Vario Micro Element Analyzer. Before analysis, the seaweed samples were washed and dried. The samples were placed in the oven at 105 °C for 3 h for the measurement of moisture content. The volatile matter was determined by putting a closed crucible containing 2 g of seaweed samples in a carbolite furnace at 950 °C for 10 min. The ash content was determined by putting a crucible containing 2 g of seaweed sample in the furnace at 850 °C for 1 h. All these amounts were assessed using the difference between the initial and final weights.

The functional group analysis of seaweed was determined using Fourier-transform infrared spectroscopy (FTIR; Operant LLC, Madison, WI, USA). The FTIR spectra were recorded in wavelengths ranging from 400 to 4000 $cm^{-1}$. Scanning electron microscopy–energy-dispersive spectroscopy (SEM-EDS) was used to confirm the microstructure of seaweed-based biochar and the elemental surface composition of the samples. The specific surface area and the pore size of seaweed biochar were assessed using a Micromeritics ASAP 2020 analyzer.

*2.3. Batch Adsorption Experiment*

Adsorption experiments of MB were performed in a glass beaker. A total of 1000 mg of MB was mixed in 1000 mL of distilled water to prepare a MB stock solution. The stock solution was later used to prepare different MB concentrations ranging from 50 to 200 mg/L. The seaweed-based biochar (300 mg) was mixed into an MB aqueous solution under continuous stirring at 300 rpm for 6 h. The solution pH was maintained in the range from 2 to 10 through dropwise addition of HCl or NaOH solutions. Some of the solution was taken every 30 min during the test. The sample was filtered using syringe filters and was analyzed for absorbance using a UV–Vis spectrophotometer (Shimadzu, Model UV 1700). The removal efficiency ($\eta$, %) and adsorption capacity of seaweed biochar were determined using Equations (1) and (2):

$$\eta = \frac{\left(C_i - C_f\right)}{C_i} \times 100 \tag{1}$$

$$q_e = \frac{\left(C_i - C_f\right)V}{m} \tag{2}$$

where $C_i$ and $C_f$ are MB concentrations (mg/L) at the initial and final stage, $q_e$ is the adsorption capacity in mg/g, m denotes the quantity of seaweed-based biochar (g), and V expresses the volume of the MB solution (L).

*2.4. Isotherm and Kinetic Analysis*

Three different temperatures of 25, 50, and 75 °C were selected for adsorption isotherm analysis tests. Table 1 specifies linearized forms of several isotherms of the Temkin, the Freundlich, and the Langmuir models [34]. For adsorption kinetic analysis, four concentrations of MB of 50, 100, 150, and 200 mg/L were selected for the pseudo-first-order and pseudo-second-order kinetic models [35], as provided in Table 1. Equations (3) and (4) were used to determine the root-mean square error (*RMSE*) and the coefficient of determination, $R^2$, to validate the model for the experimental data.

$$R^2 = 1 - \frac{\sum_{i=1}^{n} \left(q_{e,obs} - q_{e,pred}\right)^2}{\sum_{i=1}^{n} \left(q_{e,obs}\right)^2 - \left[\left(\sum_{i=1}^{n} q_{e,obs}\right)^2 / n\right]} \tag{3}$$

$$RMSE = \sqrt{\frac{1}{n-1} \quad * \quad \sum_{i=1}^{n} \left(q_{e,obs} - q_{e,pred}\right)^2} \tag{4}$$

where $q_m$ is the maximum adsorption capacity (mg/g) and $q_e$ is the adsorption capacity at the equilibrium (mg/g). The Temkin, the Freundlich, and the Langmuir constants are denoted by $K_T$, $K_F$, and $K_L$, with corresponding units of L/mg, mg/g, and L/mg. Parameter $B$ denotes the Temkin constant (J/mol). The factor n is the input data, $q_{e,\,obs}$ is the observed adsorption capability (mg/g), and $q_{e,\,pred}$ is the predicted adsorption capability (mg/g).

**Table 1.** Equilibrium and linearized equations of isotherm and kinetic models.

| Models | Equilibrium | Linearized | Ref. |
|---|---|---|---|
| Langmuir | $Q_e = \frac{q_m b C_e}{1 + b C_e}$ | $\frac{C_e}{q_e} = \frac{1}{q_m b} + \frac{C_e}{q_m}$ | [36] |
| Freundlich | $Q_e = K_f C_e^n$ | $\log q_e = Log\, K_F + \frac{1}{n} \log C_e$ | [37] |
| Temkin | $q_e = \frac{RT}{b} \ln(K_T C_e)$ | $Q_e = B\, Ln\, K_T + B\, Ln\, Ce$ | [38] |
| Pseudo-first order | $q_t = q_e\left(1 - e^{-k_1 t}\right)$ | $\ln(q_e - q_t) = lnq_e - k_1 t$ | [39] |
| Pseudo-second order | $q_t = \frac{q_e^2 k_2 t}{(1 + q_e k_2 t)}$ | $q_t = q_e\left(1 - e^{-k_1 t}\right)$ | [40] |

## 3. Results and Discussion

### 3.1. Ultimate and Proximate Analysis

Table 2 explains both the ultimate and proximate analyses of the raw *Eucheuma cottonii* seaweed and its biochar. The moisture content of the untreated raw seaweed is 7.32 wt.%, which is higher than those of its biochar (*Eucheuma Cottonii* seaweed biochar at 450 °C (BC450) and *Eucheuma Cottonii* seaweed biochar at 550 °C (BC550)). The *p*-values corresponding to the F-statistic of one-way analysis of variance (ANOVA) between moisture content, ash, and a volatile component of the pristine *Eucheuma cottonii* seaweed (PECS) and the biochar are lower than 0.05, suggesting that the heat treatment significantly lowered their contents. However, the effect of temperature is not significant when comparing BC550 and BC450, with *p*-values for all corresponding parameters of >0.05. The pyrolysis process causes dehydration of the samples in the first stage, which removes moisture [41]. The *p*-values corresponding to the F-statistic of one-way ANOVA between carbon, hydrogen, oxygen, O/C, and H/C of BC550 and BC450 are lower than 0.05, suggesting that nitrogen is purged during pyrolysis and that the absence of oxygen significantly improves biochar quality and keeps the biochar stable. The volatile matter of the raw seaweed is 53.60%, which is relatively lower than other biomass waste materials. The low content of volatile matter suggests that the seaweed is also suitable as a solid fuel, unlike other biomass waste [42]. The volatile matter drops to 18.22 for BC550 and 26.40 for BC450, which is half of the value obtained in the raw seaweed. The fixed carbon of the raw seaweed is 16.58%, while the fixed carbon for the biochar is 64.5% for BC550 and 52.47% for BC450. The higher differences in the carbon content of the raw seaweed and its biochar can be attributed to the influence of physical and chemical transformations as well as the parameters of temperature, pyrolysis time, and nitrogen gas flow rate during the pyrolysis process [43].

The ultimate analysis data show that carbon contents increased from 48.60% to 58% at a pyrolysis temperature of 450 °C and from 48.60% to 67.6% at a pyrolysis temperature of 550 °C. The variations in carbon content between the BC550 and BC450 samples demonstrate the significant impact of pyrolysis temperatures on the carbonization process. As shown in Table 2, the ultimate evaluation highlighted that the raw seaweed and its biochar showed comparatively lower H, S, and N contents along with higher C and O contents. The decrease in oxygen content at the higher temperature shows that the biochar is highly hydrophobic. The hydrophobic property of the biochar can be attributed to the formation or presence of aromatic compounds.



**Table 2.** Ultimate and proximate analyses of the pristine seaweed and seaweed biochar.

| Prepared Samples | Proximate Analysis, wt.% | | | | Ultimate Analysis, wt.% | | | | | | |
|---|---|---|---|---|---|---|---|---|---|---|---|
| | Moisture | Ash | Volatile | Fixed Carbon | C | H | N | S | O | H/C | O/C |
| PECS | 7.32 ± 0.02 | 22.50 ± 0.60 | 53.60 ± 1.3 | 16.58 ± 0.30 | 48.60 ± 0.60 | 6.94 ± 0.15 | 1.42 ± 0.08 | 0.76 ± 0.01 | 42.28 ± 1.0 | 0.14 ± 0.040 | 0.87 ± 0.030 |
| BC550 | 1.18 ± 0.04 | 16.10 ± 0.85 | 18.22 ± 1.1 | 64.5 ± 0.60 | 67.6 ± 0.45 | 5.40 ± 0.08 | 1.76 ± 0.10 | 1.24 ± 0.02 | 26 ± 0.80 | 0.08 ± 0.035 | 0.385 ± 0.014 |
| BC450 | 1.60 ± 0.06 | 19.54 ± 0.75 | 26.40 ± 1.25 | 52.47 ± 0.40 | 58 ± 0.30 | 5.75 ± 10 | 1.81 ± 0.09 | 1.36 ± 0.01 | 34 ± 1.1 | 0.099 ± 0.040 | 0.58 ± 0.014 |

PECS: Pristine *Eucheuma cottonii* seaweed, BC550: *Eucheuma Cottonii* seaweed biochar at 550 °C, BC450: *Eucheuma cottonii* seaweed biochar at 450 °C.

According to the IBI Biochar Standard [44], the optimal O/C ratio ranges from 0.40 to 0.50. This ratio keeps the lignin and maintains the quality of the biochar. An increase in the O/C ratio to 0.90 decreases the quality and stability of a biochar. On the other hand, decreasing the O/C ratio to 0.20 results in the loss of functional groups. The ratio between O and C of the raw seaweed and its biochar ranges from 0.38 to 0.87, which indicates that the biochar adsorbent is polar and contains a large number of surface functional groups, including polar oxygen. The higher the O/C ratio, the greater the number of polar functional groups of the biochar. It has been observed that these functional groups actively take part for the treatment of heavy metals as well as dyes [45].

### 3.2. Microstructure Analysis of the Seaweed-Based Biochar

Micrographs and elemental compositions of seaweed and seaweed biochar are shown in Figure 1. The pyrolysis process creates pores by releasing volatile matter and breaking the lignin content, as also demonstrated by the findings. The use of inert nitrogen in the pyrolysis process decreased the oxygen content and increased the carbon content, as shown in Figure 1b. The energy dispersive X-ray (EDX) analysis shows an increase in the carbon content from 54.6% to 69.5% and a decrease in oxygen content from 41.4% to 28.8% when comparing the raw seaweed with the resulting biochar. Pores and cavities can be observed in the seaweed biochar, which were produced during pyrolysis, as revealed in Figure 1b. Pores and voids are desirable because they increase the adsorption of MB dye by providing a higher number of adsorption active sites. The high carbon content indicates the efficacy of biochar, and the O/C ratio of 0.40 shows a medium surface hydrophobicity and stability of biochar, as also reported elsewhere [45].

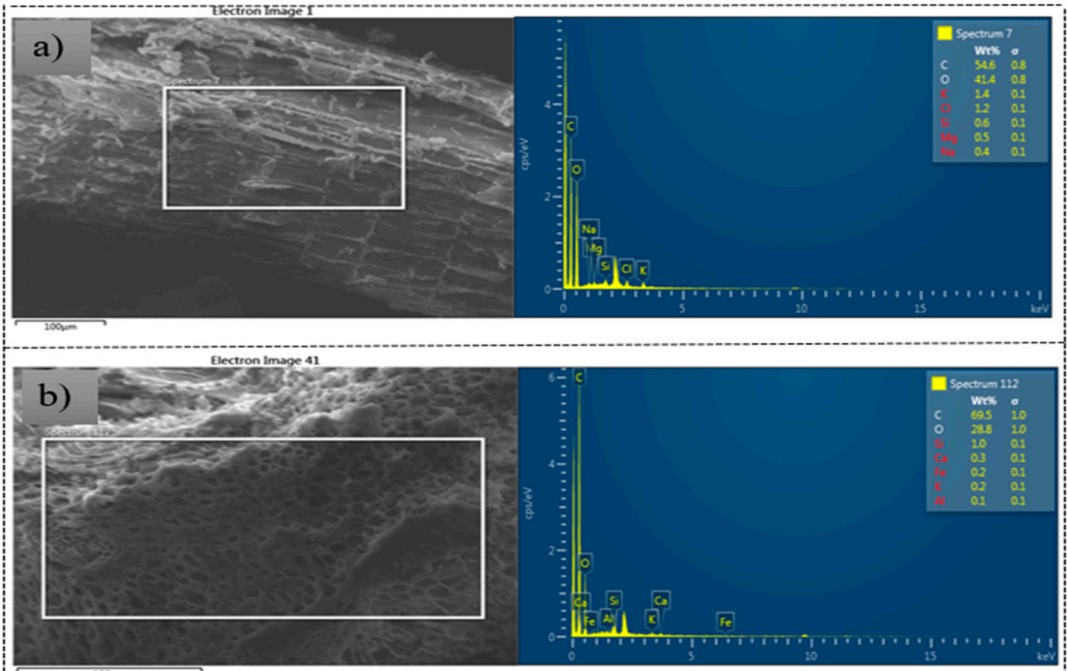

**Figure 1.** Micrographs and elemental analysis of (**a**) seaweed and (**b**) seaweed biochar.

### 3.3. BET Surface Area and Pore Size Analysis of the Seaweed-Based Biochar

Figure 2 indicates that the seaweed-based biochar can be classified as mesoporous based on the pore size distribution, according to the International Union of Pure and Applied Chemistry (IUPAC). The main adsorption isotherm resembles the typical Type-VI pattern and reflects a multi-step type of adsorption layer over a non-porous and uniform surface structure. The elevation of the steps signifies the capability of a monolayer of each and every absorption film, and in the simplest case, it is nearly constant for two or three films of absorption [46]. The $H_2$ hysteresis describes the optimal adsorption of

mesoporous solids, and the amount of nitrogen absorbed on the *y*-axis indicates that the mesoporous amounts are relatively higher. The pyrolyzed seaweed biochar has a surface area of 640.29 m$^2$/g, which is much higher than other biochar-based adsorbents extracted from agricultural and industrial biomass [47]. The seaweed biochar had a high mesopore volume of 0.54 cm$^3$/g, along with 2.32 nm of mesopore depth. The molecular structure of the MB dye is approximately 0.7 nm × 1.7 nm, which shows that the mesoporous structure and relatively higher pore volumes are compatible for adsorption of the MB [48]. Based on the observed characteristics, the seaweed-based biochar is an attractive adsorbent for MB removal.

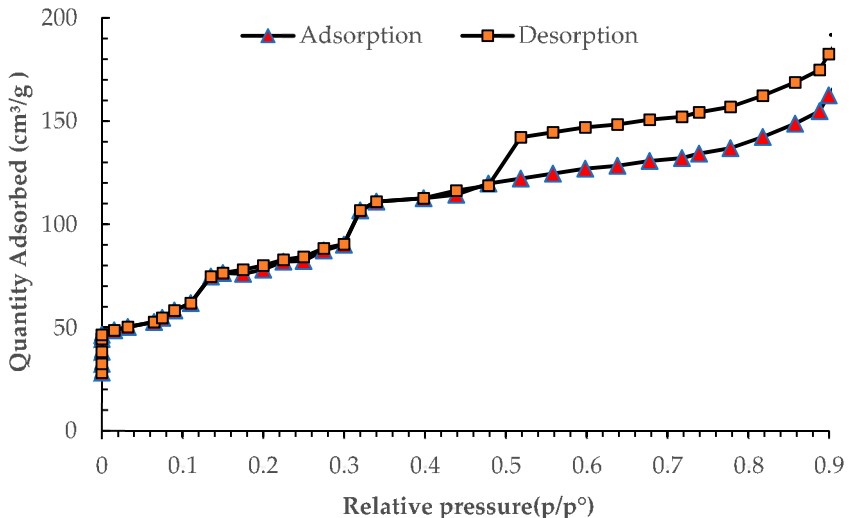

**Figure 2.** N$_2$ adsorption–desorption isotherm of seaweed biochar.

### 3.4. Functional Group Analysis of the Seaweed-Based Biochar

Figure 3 shows the FTIR spectrum of the biochar adsorbent before and after adsorption of MB dye. The peak at 3424 cm$^{-1}$, which is due to the vibration of O–H, does not change position after adsorption of MB dye [49]. The peak at 1592 cm$^{-1}$, which is a characteristic peak and caused by the stretching vibration of C=C in the biochar structure, does not show any changes after adsorption of MB [50]. In the region ranging from 1300 to 940 cm$^{-1}$, the absorption band appearing at about 1170 cm$^{-1}$ is mainly caused by the bending vibration of O–H functional groups or the stretching vibration of C–O functional groups of phenolic compounds. Another absorption peak at about 1067 cm$^{-1}$ is due to the stretching vibrations of –C–O–C– as functional groups of polysaccharides or due to –C–OH functional groups with bending vibrations [51]. O–H, N–H, and C–O functional groups are desirable for increasing the removal of MB dye.

### 3.5. Adsorption Studies of MB Using Seaweed-Based Biochar

#### 3.5.1. Influence of Initial MB Concentrations

Figure 4 shows the effect of initial concentrations of MB on the maximum removal rate. It shows that at initial MB concentrations of 50 to 200 mg/L, the adsorption activity of dye containing aqueous solutions gradually increases from 40 mg/g at an initial MB concentration of 50 mg/L to 166 mg/g at an initial MB concentration of 200 mg/L. This improvement of initial adsorption capacity could be strongly related to enhancement in the molecules of dye inside the solutions that occupy the existing active adsorption sites, resulting in the increase of adsorption activity through the employed adsorbent. The *p*-values corresponding to the F-statistic of one-way ANOVA between all the initial concentrations of MB are lower than 0.05, suggesting that the initial concentration of MB significantly influences the adsorption capacity, whereas the amount adsorbed at equilibrium for the different initial concentrations of MB (50, 100, 150, and 200 mg/g) decreases with the decrease of the initial

concentration. A comparable trend was also observed in an adsorption utilizing *Eichhornia crassipes* roots as adsorbent material for the treatment of waste effluents containing Congo Red [52]. The results show that the initial MB concentrations ranging from 50 to 200 mg/L could not saturate the biochar adsorbent, and it still has the capability to adsorb MB dye at higher concentrations.

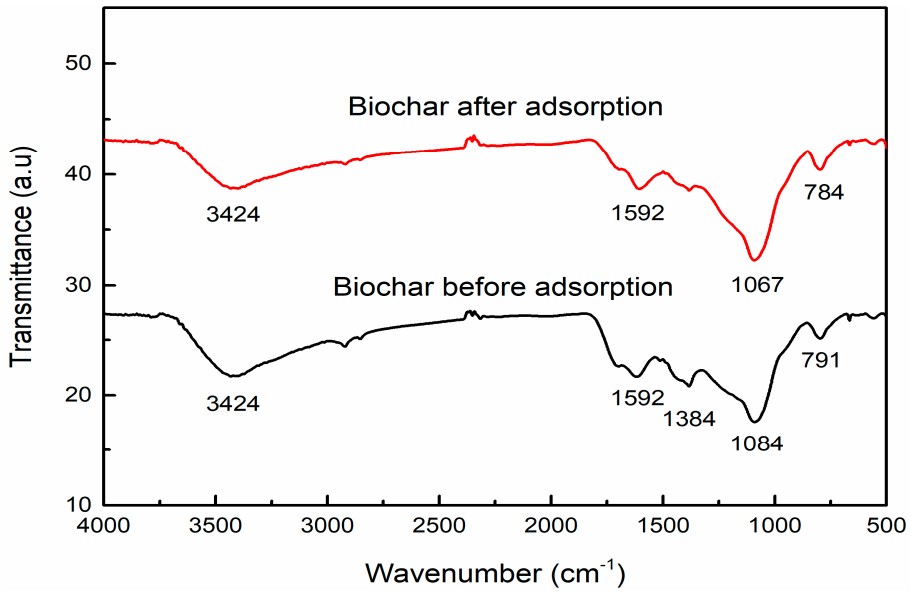

**Figure 3.** Fourier-transform infrared spectroscopy (FTIR) analysis of seaweed biochar before and after removal of methylene blue (MB) dye.

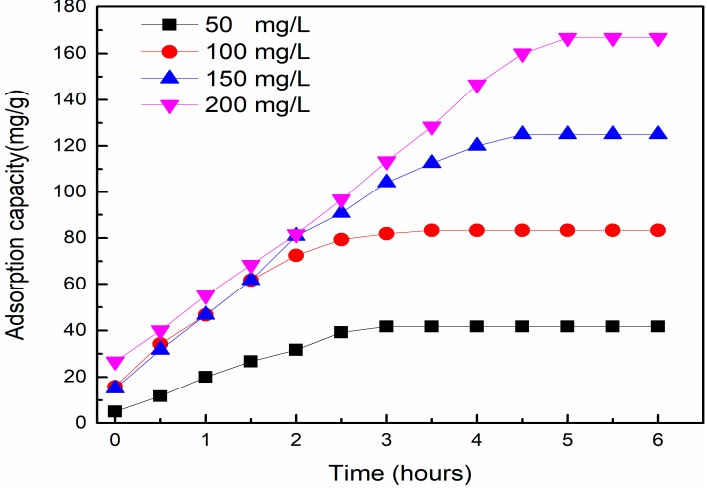

**Figure 4.** Influence of different MB concentrations on the removal activity of dye.

### 3.5.2. Influence of Solution pH

The point of zero charge (PZC) employed on the biochar adsorbent material was achieved at pH 6, as shown in Figure 5a. The PZC was determined by the titration method. This finding indicates that the biochar adsorbent becomes positively charged at pH below 6 and negatively charged at pH above 6.

Figure 5b shows the influence of pH on the adsorption of MB dye. The rise of pH values from 2 to 4 increases the removal activity of MB solutions, which ranges from 77 to 85 mg/g. Increasing the pH values further from 4 to 10 decreases the removal capability from 85 to 35 mg/g when using the developed biochar adsorbent material. The *p*-values corresponding to the F-statistic of one-way ANOVA for pH values of MB solutions are lower than 0.05, suggesting that the pH of MB aqueous solutions significantly influences the adsorption capacity, whereas the optimal pH for favorable

adsorption of methylene blue is found at pH 4. Once the pH rose to more than 4, the adsorption capacity dramatically decreased. When the pH rises above the neutral value, sedimentation of the biochar granules often occurs at the bottom of the solution, which negatively affects the adsorption system. Like the multiple reaction forces, the interaction of hydrogen ions and the spread of pores negatively affect the adsorption system. Therefore, there is a significant decrease in the elimination of selected dye pollutants from synthetic waste streams. The overall findings suggest that pH 4 is the best condition for maximum destruction of dye molecules using seaweed biochar as an adsorbent.

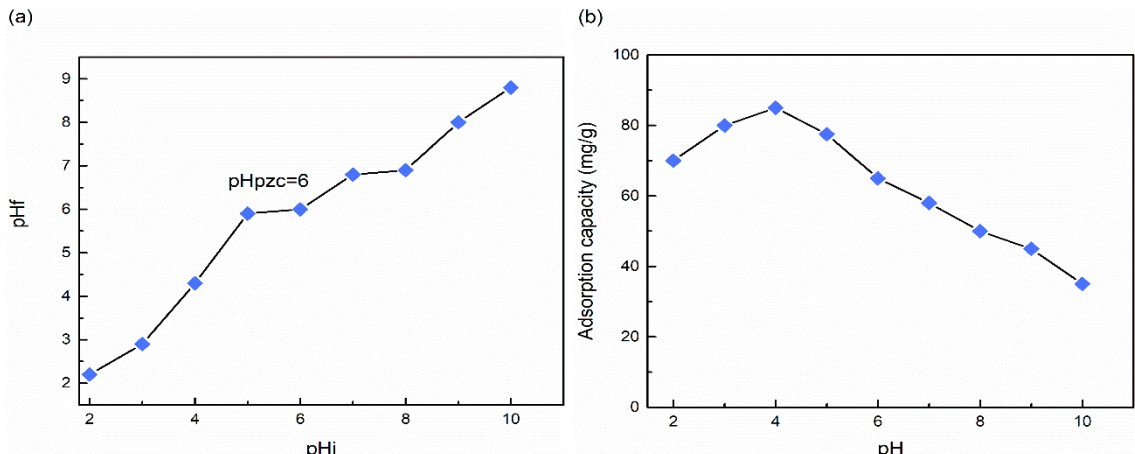

**Figure 5.** (**a**) Point of zero charge (PZC) of biochar and (**b**) influence of solution pH on maximum removal of dye.

## 3.6. Adsorption Isotherm Analysis

The adsorption isotherms of treatment of dye-containing solutions were successfully determined using experimental adsorption results by linearly fitting models of Temkin, Freundlich, and Langmuir isotherms. The adsorption isotherm graphs and results are shown in Figure 6 and Table 3. The highest values of $R^2$ of the tested isotherms suggest that the model is the most fit to the experimental data [53]. The $R^2$ values for the Langmuir, Freundlich, and Temkin models found in this study range from 0.990 to 0.994, 0.952 to 0.956, and 0.932 to 0.955, respectively. The minor differences between the experimental and suggested results attained by applying the Langmuir isotherm showed relatively smaller values of *RMSE*, ranging from 7.77 to 8.90, and these were correlated with higher values of *RSME* ranging from 9.65–15.45, which were produced by applying the Freundlich isotherm. The adsorption capacities of 107.32, 117.25, and 133.33 mg/g were obtained at 25, 50, and 75 °C, respectively, by applying the Langmuir model. The increase of adsorption capacity by increasing the temperature of the MB solution showed the endothermic nature of MB molecules' sorption with seaweed biochar. The results also show that this adsorption treatment of the selected dye with seaweed biochar occurs in the form of a monolayer. The above predictive consequences of using biochar adsorbent are well in accordance with the previously reported works on the adsorption of MB dye [10,54].

## 3.7. Adsorption Kinetic Analysis

The pseudo-first-order and pseudo-second-order kinetic model equations were applied on the adsorption data obtained from the experiments. The modeling results are shown in Figure 7 and Table 4. The highest value of $R^2$ indicates the most suitable model to explain the adsorption phenomenon [55]. A low $R^2$ of 0.940–0.988 was attained by applying the pseudo-first-order equation, while a high $R^2$ of 0.982–0.995 was found by applying the pseudo-second-order equation. On the other hand, the *RMSE* value for the pseudo-second-order equation is lower than that of the pseudo-first-order equation. The results show that the adsorption via the pseudo-second-order kinetic model is best suited for MB removal, though the pseudo-order does not fit to the data. The proposed sorption of the MB

molecules with seaweed biochar follows the chemisorption process. The kinetic model obtained in this study is consistent with other studies on the treatment of synthetic MB waste streams using N-doped microporous biochar, sewage-sludge-derived biochar, and porous biochar derived from different raw materials [16,20,56].

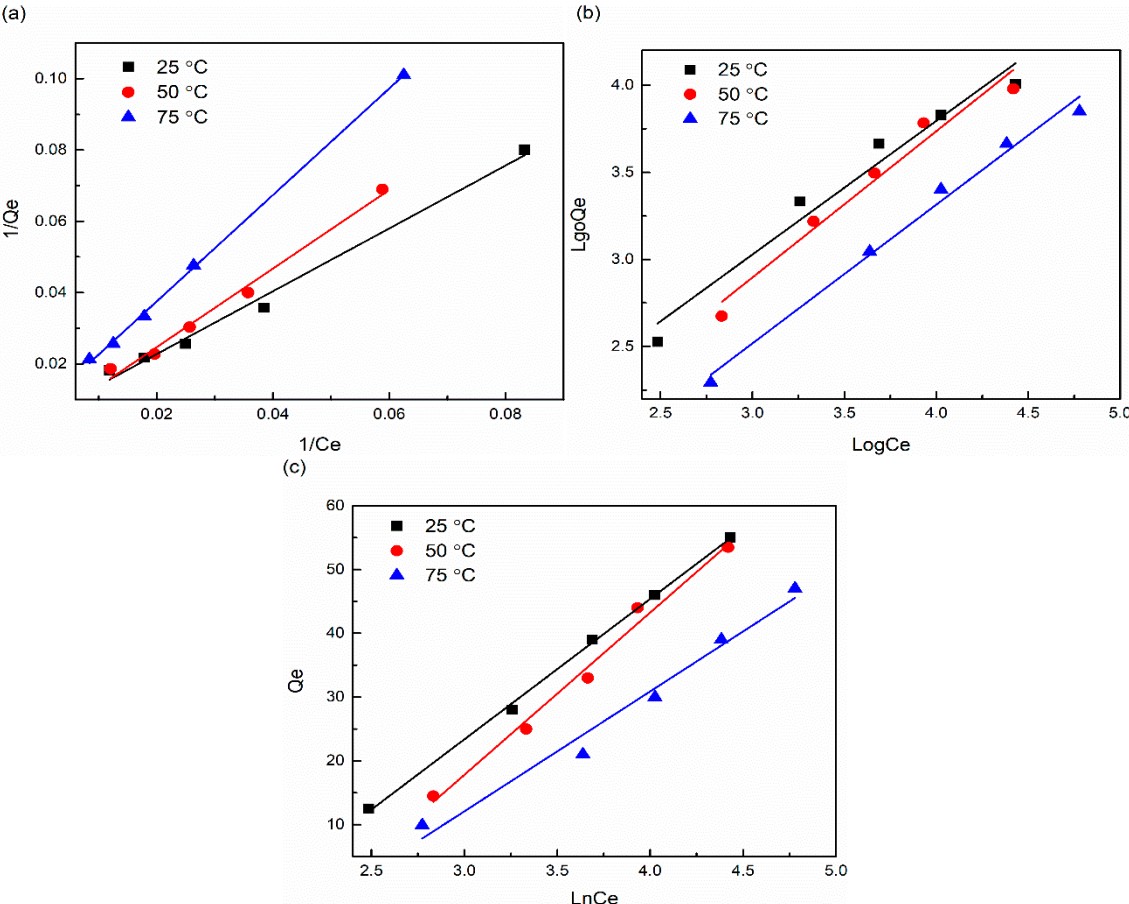

**Figure 6.** Linear fitting of various isotherm models: (**a**) Langmuir isotherm, (**b**) Freundlich isotherm, and (**c**) Temkin isotherm.

**Table 3.** Results of various isotherm parameters at three different MB solution temperatures.

| Models | Temperature (°C) | Constant | | $R^2$ | *RMSE* |
|---|---|---|---|---|---|
| Langmuir | | $q_m$ (mg g$^{-1}$) | $K_L$ (L mg$^{-1}$) | | |
| | 25 | 107.32 | 0.22 | 0.990 | 7.77 |
| | 50 | 117.25 | 0.17 | 0.992 | 8.20 |
| | 75 | 133.33 | 0.19 | 0.994 | 8.90 |
| Freundlich | | $K_F$ (L mg$^{(1-(1/n))}$/g) | $n_f$ | | |
| | 25 | 18.24 | 1.456 | 0.956 | 9.65 |
| | 50 | 38.54 | 1.438 | 0.952 | 15.54 |
| | 75 | 56.43 | 1.428 | 0.954 | 10.45 |
| Temkin | | $K_T$ (L mg$^{-1}$) | $B$ (J mol$^{-1}$) | | |
| | 25 | 0.1144 | 84.50 | 0.945 | 7.10 |
| | 50 | 0.1094 | 88.60 | 0.955 | 6.30 |
| | 75 | 0.1165 | 94.50 | 0.932 | 6.80 |

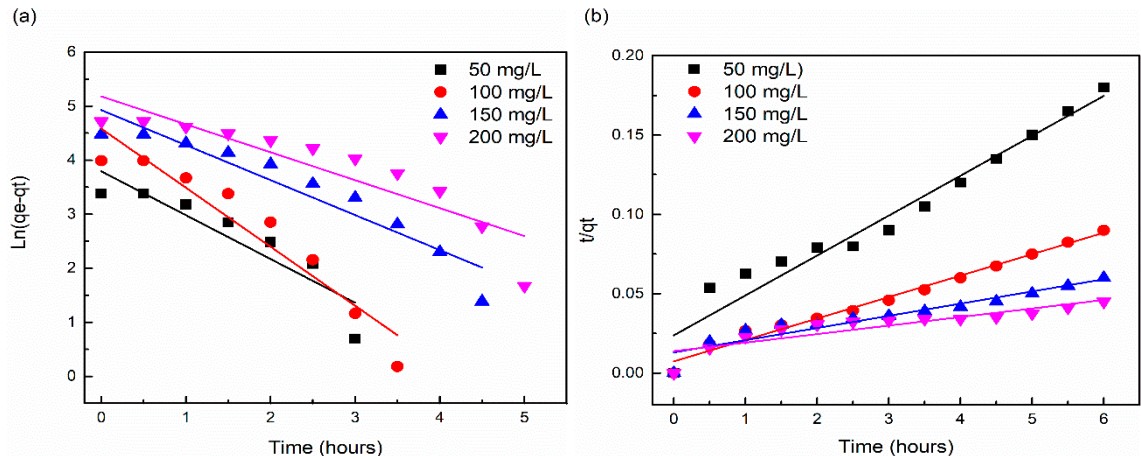

**Figure 7.** Kinetic models of the (**a**) pseudo-first order and (**b**) pseudo-second order.

**Table 4.** Adsorption kinetic parameters for the seaweed biochar–MB adsorption system.

| $C_0$ (mg/L) | $q_{exp}$ (mg/g) | Pseudo-First Order | | | | Pseudo-Second Order | | | |
|---|---|---|---|---|---|---|---|---|---|
| | | $q_{cal}$ (mg/g) | $K_1$ (h ½) | $R^2$ | *RMSE* | $q_{cal}$ (mg/g) | $K_2$ (g/mg.h) | $R^2$ | *RMSE* |
| 50 | 33.33 | 34.22 | 0.341 | 0.988 | 0.64 | 36.55 | 0.0321 | 0.995 | 0.58 |
| 100 | 66.66 | 69.34 | 0.568 | 0.966 | 1.73 | 74.55 | 0.0226 | 0.993 | 1.32 |
| 150 | 98.10 | 105.32 | 3.4 | 0.955 | 3.46 | 112.5 | 0.0131 | 0.987 | 1.91 |
| 200 | 120.67 | 132.43 | 6.4 | 0.940 | 6.92 | 138.16 | 0.0036 | 0.982 | 3.5 |

### *3.8. Performance Assessment of Biochar Compared to Literature-Based Adsorbents*

A performance comparison of the seaweed biochar with other adsorbents is shown in Table 5. The treatment activity of seaweed biochar is better than the adsorption capacities of sewage-sludge-derived biochar, anaerobic-granular-sludge-based biochar (ASG-BC), weed-based biochar (WC), and biochar from pyrolysis of wheat straw. However, the adsorption capacities of N-doped microporous biochar, magnetized Tectona Grandis sawdust, activated carbon from cashew nut shells, walnut shell powder, and oxidized weed-based biochar (OWC) are higher than that of seaweed biochar. The comparison results show that the seaweed biochar is competitive for application for the treatment of MB-containing wastewater.

**Table 5.** Performance comparison of seaweed biochar with other adsorbents for MB dye removal.

| Adsorbent | Adsorption Capacity (mg/g) | Ref. |
|---|---|---|
| N-doped microporous biochar | 436 | [16] |
| Sewage-sludge-derived biochar | 29.85 | [20] |
| Magnetized Tectona Grandis sawdust | 172.41 | [15] |
| Activated carbon from cashew nut shells | 476 | [48] |
| Anaerobic-granular-sludge-based biochar (AGS-BC) | 90.91 | [50] |
| Walnut shell powder | 178.9 | [54] |
| Oxidized weed-based biochar (OWC) | 161.29 | [57] |
| Weed-based biochar (WC) | 39.68 | [57] |
| Biochar from pyrolysis of wheat straw | 12.03 | [51] |
| Seaweed biochar | 133.33 | This study |

## 4. Conclusions

Mesoporous seaweed biochar was successfully produced as an attractive adsorbent for the removal of MB from synthetic wastewater. The optimum biochar adsorbent developed here possesses a surface

area of 640 m$^2$/g, pore volume of 0.54 cm$^3$/g, and pore size of 2.32 nm. The adsorption mechanism is of the chemisorption type, in which a single layer of MB is formed on the surface of the adsorbent. The performance of seaweed biochar is comparable to that of commercial adsorbent materials for the treatment of MB-containing synthetic wastewater.

**Author Contributions:** Conceptualization, A.A.H.S., and N.Y.H.; methodology, A.A.H.S., N.Y.H., and S.S.; formal analysis, A.A.H.S., M.R.B. and A.V.; investigation, A.A.H.S. and N.Y.H.; resources, A.A.H.S. and A.V.; data curation, A.A.H.S. and N.Y.H.; writing—original draft preparation, A.A.H.S., A.A.S., and M.Z.; writing—review and editing, A.A.H.S., A.A.-F., M.R.B. and N.A.; visualization, A.A.H.S., A.A.S.G. and N.Y.H.; supervision, N.Y.H. and S.S.; project administration, N.Y.H. and S.S.; funding acquisition, N.Y.H.; Software design, A.A.S.G. All authors have read and agreed to the published version of the manuscript.

**Funding:** This work was funded by the Ministry of Higher Education (MOHE) with the grant code FRGS-015MA0-089.

**Acknowledgments:** The authors would like to acknowledge the efforts of the Universiti Teknologi Petronas Malaysia for financing the project through the Ministry of Higher Education (MOHE) with grant code FRGS-015MA0-089.

**Conflicts of Interest:** The authors declare no conflict of interest.

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
