# Peer review of "Eucheuma cottonii Seaweed-Based Biochar for Adsorption of Methylene Blue Dye"

_sustainability, doi:10.3390/su122410318_

Round 1
Reviewer 1 Report
The work is extremely interesting and generally well reported. However, English must be extensively improved.
Lines 47-49: In this sentence, the verb is missing
Lines 62-63: 95% of what? Please explain
Line 64: Since now we are in 2020, how can the authors state that the production in 2022 was of 22 tons
Line 84: Was the seaweed pulverised before drying it? How was it possible?
Line 94: The acronym ASTM should be explained
Line 98: Is 3 hours enough? Have the authors verified if the weight remained stable after that time?
Lines 99-101: Were these procedures optimised by the authors or they were found in literature? In the latter case, references should be provided
Line 113: Was it a suspension or a solution? In the case of a suspension, it is not possible to accurately produce different concentrations from dilution
Figure 7a: It is evident from the graph that the model does not correctly interpret the experimental results. This observation should be reported in the text.
Lines 334-336: The authors cannot state that the material is suitable for the removal of other pollutants, since they have not tested them yet.
Author Response
The authors would like to thank the reviewers for their valuable comments and suggestions that help to improve our manuscript. Amendments were provided accordingly, and all modifications were highlighted red in the revised version of the manuscript. The comments' responses point by point is given in the attached file.
Please see the attached word file.

Reviewer 2 Report
The work has scientific integrity but the manuscript is poorly written. A few major issues in sentence structure and grammar are highlighted but authors are highly encouraged to thoroughly revise the entire manuscript and resubmit a well-written manuscript.
Author Response

(The authors gave the same response as above.)

Reviewer 3 Report
Authors described an interesting study about the use of seeweed derived biochar for pollutants removal.
Nonetheless, thare are some minor and capital issues:
lines 15-32: Do not use acronyms in teh abstract
line 20: Please change it as follow "The Eucheuma cottonii Seaweed biochar was produced"
line 158, table 2: All the data herein reported are provied without any associated error value. Authors must add and anlyzed the appropriated statistically tools.
line 188, figure 1: SEM scales are missed. Add them.
line 208-223: How could you say that stretching band decreased? The baselines are not sufficent clear to say it. Improved the fitting eleboration of the spectra.
The real fatal flow of thi reasearch is teh quality of the data reported in section 3.6 and 3.7. Authors did not associate with the measurments any error value. So, thewy cannot say if one data is different from the other. Measurment wihtout associated errors are just number not scientific value.
Accordingly with the concerns raised above, i discourage the pubblication of this work as it is in this and in all reputated journal.
It is not reached the consistency level requiered for a reproducible pubblication.
Author Response

(The authors gave the same response as above.)

Round 2
Reviewer 2 Report
The manuscript has significantly improved however, minor spell and grammar checks are required.
Author Response
Response to Editor and Reviewers
The authors would like to thank the reviewers for the valuable comments and suggestions that contribute to improve our manuscript. Amendments were provided accordingly, and all modifications were highlighted red in the revised version of the manuscript. Response to the comments is given below.
Reviewer 2
Comments and Suggestions for Authors: The manuscript has significantly improved; however, minor spell and grammar checks are required.
Response: Thank you, the manuscript has been Grammarly corrected and revised, please see the revised manuscript.
See the attached word file

Reviewer 3 Report
Authors still misses the stastically analysis of the data. Add the errors is fine but it is required to analyzed the data by using statistically instruments such as ANOVA. Otherwise how could you say that one is different from the other?
They did not get the point of adding error and deeply analyze thier data
Author Response
Response to Editor and Reviewers
The authors would like to thank the reviewers for the valuable comments and suggestions that contribute to improving our manuscript. Amendments were provided accordingly, and all modifications were highlighted red in the revised version of the manuscript. Response to each comment, point by point, is given below.
Reviewer 3
Comments and Suggestions for Authors: Authors still misses the statically analysis of the data. Add the errors is fine but it is required to analysed the data by using statistically instruments such as ANOVA. Otherwise, how could you say that one is different from the other? They did not get the point of adding error and deeply analyze their data
Response: Thank you for the valuable comments. The authors added the ANOVA analysis description where needed as suggested. Please see lines 145-153, lines 233-237, and lines 258-261 in the revised manuscript.
See the attached word file.
